# Lessons Learned from Age-Friendly, Team-Based Training

**DOI:** 10.3390/geriatrics8040078

**Published:** 2023-07-28

**Authors:** Sara C. Murphy, Jennifer J. Severance, Kathlene Camp, Janice A. Knebl, Thomas J. Fairchild, Isabel Soto

**Affiliations:** Department of Internal Medicine and Geriatric, University of North Texas Health Science Center, Fort Worth, TX 76107, USA

**Keywords:** geriatrics, age-friendly, healthcare improvement, leadership, interprofessional collaboration

## Abstract

According to the Institute of Medicine, immediate steps must be taken across the United States to educate and train the healthcare workforce to work collaboratively to address the needs of the growing older adult population. The Geriatric Practice Leadership Institute (GPLI) was designed to support professional teams working in acute and post-acute care in transforming their organization into a designated Age-Friendly Health System. The program was built around the Institute for Healthcare Improvement’s Age-Friendly Health Systems 4Ms framework. This framework focuses on What Matters, Medication, Mentation, and Mobility (the 4Ms) in supporting care for older adults. The GPLI program is an online, seven-month team-based program with four to seven participants from one organization per team. Additionally, each team selected, developed, and completed a quality improvement project based on Age-Friendly Health Systems 4Ms. The curriculum also includes organizational culture, leadership, and interprofessional team-building modules. Using a post-completion survey, the experiences of 41 participants in the GPLI program were assessed. All respondents found the information in the program ‘very’ or ‘extremely’ valuable, and their executive sponsor ‘very’ or ‘extremely’ valuable in supporting their team’s involvement and project. The GPLI program has trained over 200 healthcare professionals and teams that have successfully implemented projects across their organizations.

## 1. Introduction

By 2030, the number of adults aged 65 years and over will double to 24% of the United States (U.S.) population [1,2]. The Texas older adult population, the third largest in the U.S., is increasing faster than that of the nation, presenting significant challenges to current and future healthcare delivery [3,4]. In addition, many older adults have multiple chronic conditions, including diabetes, hypertension, heart disease, and Alzheimer’s Disease and Related Dementias (ADRD) [5]. National data also show that one in three older adults falls each year, presenting another public health and safety concern, resulting in more than USD 31 billion in annual Medicare costs [6]. Furthermore, over two million U.S. citizens aged 65 and older suffer from some form of depression, and mental health issues exacerbate chronic illnesses [7]. Chronic care management is complicated by social and environmental factors, such as income insufficiency, social isolation, or low health literacy, which can negatively affect a provider’s ability to influence health outcomes [8]. With two out of three older adults with multiple chronic conditions, the aging population demands broad support from integrated care systems. Managing complex chronic diseases is a collaborative task that addresses various physical and mental risk factors that impact health outcomes. 

Given the increase in the older adult population and the complexity of care, older adults are at a higher risk of healthcare-related harm due to increased healthcare utilization. The focus on healthy aging has received national and global attention in recent years and has been integrated into academia and healthcare delivery, including the United Nation’s Decade of Healthy Aging: Plan of Action 2021–2039 [9,10]. According to the Institute of Medicine, immediate steps must be taken across the U.S. to educate and train both the current and future healthcare workforce to act collaboratively in addressing the diverse needs of the growing older adult population as part of the WHO Healthy Aging Goals [10,11]. However, most healthcare professionals traditionally receive minimal exposure to geriatric content in their education [12]. Considering these factors, it is evident that healthcare providers working with older adults should be provided with the appropriate knowledge and skills to care for older adults and ensure better patient outcomes. Therefore, it is incumbent upon the healthcare system to give providers geriatric training focused on working successfully as a clinical team in caring for older adults to improve patients’ quality of life. Unfortunately, so far, geriatric education and training as well as workplace organization fail to embrace a team approach [13]. Additionally, healthcare providers need to be able to lead change within their organizations to provide the age-friendly care that older adults need.

The increased demand for primary care is most prominently seen by general/family medicine and general internal medicine practitioners. Moreover, based on the latest projections of the Health Resource Service Administration (HRSA), the national demand for primary care physicians is projected to increase from 224,780 in 2013 to 263,100 in 2025, a 17% increase [14]. Unfortunately, as the older adult population grows, organizations such as the Association of American Medical Colleges forecast a physician shortage of 54,000 to 139,000 by 2033 [14]. To make matters worse, the COVID-19 pandemic has stressed clinical providers’ physical and emotional health, leading to burnout and early retirement. In particular, clinical providers in nursing homes are disproportionally affected by COVID-19 and the lack of preparedness in that setting due to the nature of the vulnerable population they care for [15]. 

These factors have resulted in healthcare systems needing help to provide evidence-based practice to every older adult at every point of contact in the healthcare system. In addition, as the U.S. population ages, our healthcare system must be better prepared to care for older adults. The Age-Friendly Health System 4Ms framework, developed by the John A. Hartford Foundation, refers to the effort to use evidence-based elements of high-quality care for older adults [16,17]. In the 4Ms framework, What Matters means to know and act on an older adult’s health outcome goals and care preferences for current and future care, including end-of-life; Medication means that if medication is necessary, use age-friendly medication that does not interfere with What Matters, Mentation, or Mobility; Mentation means to prevent, identify, treat, and manage dementia, delirium, and depression; Mobility means to ensure that older adults move safely every day in order to maintain function and do What Matters [18,19]. Focusing on the core concepts, the 4Ms allow for a more manageable treatment approach. Moreover, the 4Ms framework provides care centered around a patient’s overall well-being instead of a disease state, which applies to all geriatric care patients. An expert panel has summarized the literature and reviewed the evidence for the 4Ms. The panel found that the evidence for the 4Ms is robust, and that it creates an elegant way to ensure that older adults reliably receive the best care possible [19]. The team-based Geriatric Practice Leadership Institute (GPLI) program was created to address the growing challenges in caring for older adults through training healthcare teams based on the Age-Friendly Health System 4Ms framework [16]. In this paper, we explain the purpose and content of the GPLI program and present the results of a post-completion survey of GPLI program participants. As an example, we present the results of a project of one of the GPLI teams. 

## 2. Materials and Methods

The team-based GPLI program is a tuition-free program formed from a shared vision initiated in 2013 by geriatric clinicians from the University of North Texas Health Science Center (UNTHSC) and executive education leaders from the Texas Christian University (TCU) Neeley School of Business to inspire and promote transformational team-based solutions that optimize outcomes for older adults. The program has been funded by the HRSA since 2013, and has been offered to over 200 participants from acute and post-acute organizations. 

The GPLI program aims to provide participants with a solid foundation to begin their organizational journey to becoming recognized as a Level 1 Age-Friendly Health System or continue their age-friendly journey to Level 2 if Level 1 designation has already been achieved. The program prepares early- and mid-career professionals working in teams to become clinical leaders in their organizations. In addition, the GPLI program provides participants with skills and knowledge to improve patient care for older adults by incorporating the Age-Friendly Health Systems 4Ms framework. The GPLI shifted to be based entirely on the Age-Friendly Health System 4Ms framework three years ago, and has now graduated from two cohorts of teams from an age-friendly program. Two years of graduates from this age-friendly program were asked to complete a survey to assess the program’s success at program completion. The survey consisted of two open-ended questions and six closed-ended five-point Likert scale questions, where one equaled ‘Not at all valuable’ and five equaled ‘Extremely valuable’. 

### GPLI Program Components

The teams comprised four to seven participants and could be pre-existing or assembled for the GPLI program. Individuals from various disciplines participated. Previous teams included nurses, physicians, social workers, pharmacists, rehabilitation therapists, administrators, regional managers, C-suite executives, and social service providers. Each team was assigned a coach with expertise in healthcare delivery and business leadership, who worked with the team for the duration of the program, offering professional support and guidance throughout the project development and implementation process. Coaches who had previously implemented the age-friendly model were available to assist teams with concerns about integrating the model into their quality improvement projects. Coaches regularly met with teams to provide support based on the participants’ learning needs and GPLI team members’ priorities. Additionally, each team had an executive sponsor to help with project sustainability, potential funding needs, navigating organizational culture, and cultivating changes in the organization’s ecosystem. The projects covered topics ranging from fall prevention and behavioral management to nonpharmacologic interventions in nursing homes. 

The participants completed five online, self-paced sessions and associated individual and team writing assignments, as outlined in Table 1 (for a detailed description of the GPLI program, see Appendix A). In addition to using the Age-Friendly Health Systems: Guide to Using the 4Ms in the Care of Older Adults and other Institute of Health Improvement resources, teams used several Harvard Business Review articles, Kotter’s (2012) framework for change management, and Strengths Based Leadership [20]. Each group also completed a Quality Improvement project or project plan based on the Age-Friendly Health Systems 4Ms framework and a final project report and presentation. In addition to the asynchronous online curriculum, participants attended three obligatory one-hour Zoom class meetings to share the lessons learned. The modules were sequenced to provide participants with an overview of why organizational culture and individual leadership are critical foundational elements for achieving success in the Age-Friendly journey, before diving into content details on quality improvement and the Age-Friendly Health System 4Ms framework. Participants were offered continuing education credits and a micro-credential to encourage their participation.

## 3. Results

A total of 41 participants, consisting of nine project teams over two years, participated in the study, with a response rate of 73%. The program participants ranged in age from 20 to 60, were mostly female (76%), and had a range of backgrounds in internal medicine, clinical social work, nursing, emergency medical services, pharmacy, electronic medical record administration, healthcare administration, and academia. In the post-completion survey (see Table 2), all respondents rated the content of the program as either ‘very’ or ‘extremely’ valuable. Similarly, all respondents found their executive sponsors to be either ‘very’ or ‘extremely’ valuable in supporting their team’s involvement and project. Positive ratings included applicability to current positions, usefulness of skills learned, relevance to future career goals, executive sponsor supportiveness, and coach effectiveness. 

It was evident from the open-ended queries that participants became agents of change within the health system while also increasing their knowledge of Age-Friendly Health Systems. The participants were instructed by their team coach on how highly effective teams function and how to set and achieve team objectives, which are crucial when caring for older adults. In addition, the GPLI program helped participants improve their use of effective strategies to incorporate evidence-based practices into their existing care. The program raised awareness of how enhancing patient-centered geriatric care, patient safety, and workflows can positively affect older individuals’ health outcomes and quality of life. This newfound awareness empowered participants and teams to spearhead their organizations’ preparations for Age-Friendly Health System certification. Two organizations have already earned this distinction, and two others have pending applications.

### 3.1. Example GPLI Team

In 2021, MedStar, a local government agency that provides a wide range of services, including acute emergency medical response, flu vaccinations, and a mobile integrated health (MIH) program, teamed up with the Alzheimer’s Association and UNTHSC SaferCare Texas, supported by a team coach from the TCU Neeley School of Business. Their GPLI project involved finding effective ways to integrate the 4Ms framework into the emergency medical services (EMS) system through emergent responses and MIH services. This initiative was built upon a prior MedStar MIH GPLI team initiative to integrate fall risk assessment and intervention into the current in-home comorbidity management for high-utilization callers. In addition, the emergency responder team sought to provide what matters, medication, mentation, and mobility interventions to the MedStar 911 calls. 

In addressing what matters, the question ‘What matters to you today?’ with guided categories including family, health, and religion/spirituality was added to the electronic medical record (EMR) algorithm for paramedics. Additionally, 911 staff members were coached to ask the patients to determine the course of action. Considering the complexities of treating older adults, addressing what matters in an emergency could avoid unwanted emergency room visits or hospitalizations and focus instead on quality-of-life goals. 

Regarding medications, as older patients are commonly on at least one prescription drug, addressing medications in an emergency could avoid drug mismanagement or adverse drug events. To address medication, any reported medications on the Beers’ list, an established list of potentially inappropriate medications for older adults, were automatically set to flag in the EMR [21]. This generated a warning message prompting paramedics to verify the potential risks of flagged medications and to recommend a follow-up visit with the patient’s physician or pharmacist, if needed. Embedding these interventions into the EMR ensured their use by MedStar providers each time the patient was evaluated. 

As for mobility, falls as a significant contributor to emergency calls, screening, and addressing fall risk factors could significantly reduce subsequent calls or medical care and associated morbidity and mortality from fall-related injuries. To address mobility, an inquiry using the Centers for Disease Control and Prevention (CDC) Stopping Elderly Accidents, Deaths & Injuries (STEADI) Stay Independent Investigation was also embedded into the EMR [22]. Answers to inquiries can be proxied by close family members or friends. In addition, a scan of the home environment looked for common hazards within the home or entry, which could increase the fall risk. 

Dementia, depression, and delirium negatively impact the management of health conditions and lead to a higher fall risk. Screening mentation provides earlier awareness and interventions to remediate or control negative health outcomes. The 911 responders were asked four questions to screen for dementia. 

Some patients were enrolled in the MIH program as a follow-up to the EMS visit. This program expanded the use of the 4Ms framework to further assess and act on areas needed to manage health conditions, decrease fall risk, and provide targeted care intervention, often serving as a liaison between the patient and the required providers and community services. In expanding on what matters, paramedics addressed subjects related to advance care directives, patient goals, and what matters most to support their quality of life. Moreover, during in-home MIH visits, managing what matters provides an opportunity to align care with the patient’s priorities. Further education and discussion with the patient, caregivers, and their medical providers helped address the high-risk medications flagged in the EMR. Regarding mentation, screening tools, including the Montreal Cognitive Assessment (MoCA), Ascertain Dementia (AD8), Patient Health Questionnaire (PHQ-9), and Geriatric Depression Scale (GDS), were used to screen for dementia and depression, thus guiding appropriate community services and support [23,24,25,26]. Paramedics then employ the Timed Up and Go (TUG) test to assess mobility and fall risk and review home-safe modification needs. Connections between physical therapy providers and community services were initiated to address these needs. 

### 3.2. Outcomes of the GPLI Project

MedStar still uses the 4Ms framework to identify factors impacting health management and emergent needs and aligns findings with patient-centered and targeted needs by coordinating care with physicians and community services. As of January 2023, MedStar continues to use what matters in its EMR. In September 2021, an in-person what matters continuing education session took place with the entire MedStar staff, emphasizing the idea and its implication in the care of older adults. Another in-person training was conducted, which focused on communication with the emergency department staff. For medication, MedStar 911 and MIH paramedics continue to flag Beers’ list of medications in their EMR when obtaining patients’ medical histories. Additionally, paramedics verify the potential risks and develop care plans based on the medications taken. Through these actions, MedStar demonstrated efficient and effective execution of the 4Ms framework. In 2022, the 911 team screened 25,912 older adults for risk factors affecting falls and health management. A total of 775 referrals from 911 calls received follow-up care from the MIH team.

Several organizations participating in the GPLI program have already been designated as Age-Friendly Health Systems. For example, the Family Medicine Clinic at the UNTHSC included a physician’s assistant, clinic director, licensed social worker, and medical assistant on the GPLI team. The UNTHSC Family Medicine Clinic has a large population of older adults with Medicare or dual eligibility that requires additional attention to ensure excellent care. The clinic sought to become a designated Age-Friendly Health System to meet the needs of this population and better serve older patients by connecting them to appropriate resources. The clinic focused on integrating 4Ms into the clinical workflow for annual wellness visits. The team added additional eligibility criteria to the electronic check-in flow and created a template for the annual wellness visit that included the 4Ms. As a result, the team increased their patients’ annual wellness visits by 84% over the two years.

## 4. Discussion

The results of the post-completion survey among GPLI program participants demonstrated the program’s positive impact on healthcare providers working with older adults. The GPLI program successfully equipped participants with the knowledge and skills of the Age-Friendly Health Systems 4Ms framework needed to care for the growing older adult population, effectively addressing the diverse needs of this population. In addition, the Age-Friendly Health System 4Ms framework has played a pivotal role in enhancing the care provided to older adults by focusing on patients’ overall well-being rather than solely on their disease states.

The collaborative team-based approach of the GPLI program has fostered an environment in which professionals from various disciplines work together to improve care for older adults. This approach has resulted in the development of highly effective teams to set and achieve the objectives crucial for caring for older adults. Moreover, the program has facilitated the integration of evidence-based practices into existing care systems, leading to improvements in patient-centered geriatric care, patient safety, and enhanced workflows.

The GPLI program empowered healthcare providers to spearhead their organizations’ preparations for Age-Friendly Health System certification and contributed to the successful designation of several participating organizations as Age-Friendly Health Systems. The example of the UNTHSC Family Medicine Clinic demonstrates how implementing the 4Ms framework can lead to significant improvements in patient care, such as increasing annual wellness visits by 84% over two years.

One key aspect of the GPLI program that has contributed to its success is the inclusion of executive sponsors and coaches in the program structure. Executive sponsors provide essential support for navigating the organizational culture, securing funding, and promoting sustainability. In addition, coaches with expertise in healthcare delivery and business leadership guided the teams throughout the project development and implementation process. This support system facilitated the integration of the Age-Friendly Health System 4Ms framework into the daily practices of healthcare providers and promoted lasting changes within their organizations. Based on feedback from the team coaches, program faculty, and participants in the program, the following lessons learned have been identified.

### 4.1. Team Representation

The teams most successful in achieving change in their organizations had representatives from all levels and disciplines where the quality project was implemented. Additionally, as highlighted by the post-completion survey, executive sponsor support is critical to individuals’ success, teamwork, and development. Feedback from all patient-facing team members was crucial for implementing and, more importantly, sustaining age-friendly changes. Recognizing that care for older adults involves many healthcare providers with various backgrounds and formal training is critical to creating an age-friendly environment within medical practice.

### 4.2. Culture Change

Quality improvement is important. While quality improvement tools can seem complicated at first glance, they become simple to execute when individuals are trained on how to use them. Change management, however, is a more complex part of the transition to providing age-friendly care that requires consistent, intentional effort. Finding local champions and early adopters, especially physicians or other leaders, is critical to seeing participation in and commitment to the program. Finding barriers in small tests of change was good, and teams that adopted this mindset were successful. Changing a culture takes time, anywhere from three to five years, to create a long-lasting positive change that will support all the elements of an Age-Friendly Health System. Education is essential to age-friendly culture change but is not always immediately successful. It takes time and a trusting relationship to help team members and patients understand the importance of an Age-Friendly Health System and the 4Ms. As exemplified by the MedStar team, continued training and education are important to effectively employ the 4Ms framework.

### 4.3. Flexibility

Staffing has been challenging for teams across settings and has been exacerbated during the COVID-19 pandemic. Setting reasonable expectations that hold teams accountable while allowing flexibility helps them deal with changing work demands. The most successful teams selected something small that was obtainable to help build momentum. Additionally, resilience when facing challenges is essential for leaders working to change their culture.

### 4.4. Limitations and Future Directions

Despite the success of the GPLI program, some limitations should be considered. First, the impact of the program may be limited by the small number of participants and the focus on a specific geographical region. Expanding the program to include more participants and healthcare providers from various regions would allow for a broader understanding of the program’s effectiveness in diverse settings. Second, the survey assessing the program’s success may not capture the full range of outcomes and challenges that participants face while implementing the 4Ms framework. Third, the age-friendly curriculum has only been part of the program for two years, so the conclusions are based on a short time span. Future research could explore the long-term impact of the GPLI program on healthcare providers, organizations, and patient outcomes through a more comprehensive evaluation methodology, including qualitative interviews and longitudinal follow-ups. In addition, as the growing older population and the associated challenges in healthcare are a worldwide problem, with the world’s population aged 60 years and older expected to increase by 1.4 billion by 2030, future directions should consider a roll-out of similar training in other countries as well [25].

## 5. Conclusions

The GPLI program effectively provided healthcare providers with geriatric training focused on the Age-Friendly Health System 4Ms framework. As the older adult population grows, it is crucial to equip healthcare providers with appropriate knowledge and skills to ensure better patient outcomes. The success of the GPLI program highlights the importance of team-based collaborative approaches in managing complex chronic diseases and improving the quality of life of older adults. Furthermore, including executive sponsors and coaches in the program structure emphasizes the value of organizational support in promoting lasting changes in healthcare systems. As the demand for primary care physicians and healthcare providers skilled in geriatric care continues to rise, programs such as the GPLI will become increasingly essential for addressing the complex healthcare needs of the aging population.

## Figures and Tables

**Table 1 geriatrics-08-00078-t001:** GPLI Program Topics.

Topic	Assignment
Orientation	N/A
The iceberg of culture and strength-based leadership	StrengthFinder, Kotter’s planning for change and team
Leadership and teams	Your leadership journey and how you are perceived at work
Age-Friendly Health Systems	Health system background overview
Quality improvement in Age-Friendly Health Systems	Final team charter
Putting the age-friendly model into practice	Age-friendly appendix c, PDSA Worksheet

Content and assignment from the Age-Friendly GPLI Program.

**Table 2 geriatrics-08-00078-t002:** GPLI program post-survey responses (N = 30).

Question	Not at All Valuable	Slightly Valuable	Moderately Valuable	Very Valuable	Extremely Valuable
How valuable was the module information you received in GPLI for your current position?	0	0	0	16 (53%)	14 (47%)
How valuable were the skills you learned in GPLI for your current position?	0	0	0	16 (53%)	14 (47%)
How valuable will the module information you received in GPLI be for your future career goals?	0	0	0	14 (47%)	16 (53%)
How valuable will the skills you learned in GPLI be for your future career goals?	0	0	0	14 (47%)	16 (53%)
How valuable was your executive sponsor in supporting your team’s involvement/project in GPLI?	0	0	0	14 (47%)	16 (53%)
How valuable was your coach in your GPLI activities?	0	0	2	13 (46%)	15 (54%)

GPLI survey questions, answer choices, and responses.

## Data Availability

Data are available upon request.

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
