# Peer review of "Lessons Learned from Age-Friendly, Team-Based Training"

_geriatrics, 2023, doi:10.3390/geriatrics8040078_

Round 1

Reviewer 1 Report

 The present work refers to the lessons learned from an age-friendly and team-based training program, established in US, and it could be of great general interest as part of the Decade of healthy aging goals. However, the program is not presented per se, so the eligibility to refer to lessons is flattened. In the previous review process (submission), as documented in the supplementary data, the reviewers already detected this gap as the weakest point. Authors amended the Ms ‘removing language where measurement was not present’. There’s no data nor checklist items of what it is measured. Still, because of the general interest such a program can have, in order to consider this work as a potential scientific article,  it is a must to ‘describe’ the program protocol, 2) the contents (Table 1 refers to topics, not to contents, they can’t be referred as ‘information in the program’ given by the in line 139). In a certain way, these two points would refer to what I can expect to find if I search about such a program online and I make decision to enrol or not based on the program and contents described in the website. A 3) question are to describe the variables measured in the survey and in the report of results to provide specific data/report on which ‘content’ (information) is rated as very valuable, which extremely valuable, etc. Similarly, which knowledge, which skills (lines 240) etc?  Some figures would help to illustrate the variables measured. 

Abstract and introduction. In order to be of general interest, please, note that the authors attribute to an institution (Institute of Medicine) a must/request for US, that in fact, applies to most worldwide population and is part of WHO Decade of healthy aging goals. I strongly suggest including the temporal frame of the Decade, and the worldwide population statistics, so the Ms is of broader interest.

After reading the previous reviewers' comments and queries, I strongly recommend providing a scientific and specific report to the question “how can you tell it is effective?” rather than a general (no specific) answer such as  “In conclusion, the GPLI program effectively addressed the challenges of caring for the growing older adult population by providing healthcare providers”

In summary, it has a great potential, but needs to be specific and to trust the good considerations, the readers need to see a scientific report. I encourage the authors to do so, as it is clear that their experience with the program has been positive and it is worth to share such interventions.

Reviewer 2 Report

You created a evaluation of a program instituted for seniors, and their health care. This is basically an evaluation of a program by the participants concerning their involvement and opinions of this particular program. Overall it seems rather positive, and nothing on the critical or improvement side, i would suggest a question concerning, what did you not like, how would you improve the classes, etc. For so many authors, you seem to have a rather small number of references. 

level of english was very good, i only saw one or two mistakes

Reviewer 3 Report

The authors have engaged in a project that has the potential to benefit the medical care provided to older adults, Unfortunately, this manuscript falls short in providing useful information to readers interested in building geriatric teams that support older patients. That is, the authors have provided more of a summary of  their efforts rather than details and insights to enable others to replicate or build upon their experiences. I strongly suggest the authors rethink how they present the material and their findings in the manuscript. For example,

The introduction provides rationale for coordinated team care, but that is unnecessary. Rather, build on the the knowledge that geriatric education and training and workplace organization fails to embrace a team approach. Provide some background on that (with citations). Then introduce the 4M Framework. Discuss each "M" and provide some background support for the development of and scientific evaluation of the 4M Framework and why it is a reliable framework to replicate. Provide lots of scientific citations.

As the authors noted, the use of post-training data as a measure of the effectiveness of a training is quite limited. Afterall, people will often report their experience as "excellent" immediately following a training. If the authors have no other meaningful training or satisfaction data, perhaps they can draw on trainer or mentor experiences and how they interpreted the successes and challenges of their teams. This would help inform how reader's might apply your findings to their organizations.

Lastly, in keeping with this above restructuring, write about potential changes to the 4M training you provided,  next steps for institutionalizing the 4M framework across departments within organizations. and any other tips for implementation people interested in implementing the framework need to know.

By reframing the current  manuscript, the information within will have a larger impact by reaching a wider audience.

Reviewer 4 Report

Thank you for the opportunity to review this manuscript that showcases a very interesting age-friendly program.

I have a comment and few minor edits suggestions

The Abstract (lines 11-12)  could include more context about the Institute for Healthcare Improvement's Age-Friendly Health Systems 4Ms framework as this might not be familiar to a reader outside the USA.

(line 59) In keeping with an 'age-friendly' discourse, I'd recommend to  change 'geriatric population' for 'older adults population'

Also 'geriatric patients' for 'geriatric care patients' (line 74) and the term 'Americans' for 'US citizens' (line 33)

Round 2

Reviewer 1 Report

The authors have addressed all the issues raised by this reviewer in their rebuttal letter, that were mostly related to the lack of specific report on the program contents and features, and also the need to put the efforts and achievements in context of the decade of healthy aging. The changes have been clearly stated and also highligted in the text using red colour. The new supplementary data and these amendments have resulted in a new revised version that provides a more descriptive scenario, and contextualization, that can be more useful for those interested in the field, and be a good guideline for others of a succesful program design and implementation.

I'd like to thank the authors for their sensitivity towards this critical issue, and for sharing their knowledge in an article to help other professionals and to benefit the older people, who are the final beneficiaries.

Reviewer 3 Report

The second revision of the paper improves the quality of the manuscript. However, it remains that the manuscript is basically a report, and not written as a scientific study nor a scientific program or curriculum evaluation. The citations for the 4M work are only descriptive studies and reports from the original research team. Identifying outside scientific studies would certainly strengthen the rationale for using the framework and support the evidence of findings. The authors continue to assume that the reader will accept at face value that an "expert panel" created the framework and their conclusion that the training successfully equipped participants to care for their patients without data other than self-report.  I want to like this manuscript, but there are too many gaps of information.